

# Universal and non-universal large deviations in critical systems

Ivan Balog[1⋆], Bertrand Delamotte[2] and Adam Rançon[3†]

**1** Institute of Physics, Bijenička cesta 46, HR-10001 Zagreb, Croatia
**2** Sorbonne Université, CNRS, Laboratoire de Physique Théorique de la Matière Condensée,
LPTMC, F-75005 Paris, France
**3** Univ. Lille, CNRS, UMR 8523 – PhLAM – Laboratoire de Physique
des Lasers Atomes et Molécules, F-59000 Lille, France

⋆ balog@ifs.hr , † adam.rancon@univ-lille.fr

## Abstract

Rare events play a crucial role in understanding complex systems. Characterizing and analyzing them in scale-invariant situations is challenging due to strong correlations. In this work, we focus on characterizing the tails of probability distribution functions (PDFs) for these systems. Using a variety of methods, perturbation theory, functional renormalization group, hierarchical models, large $n$ limit, and Monte Carlo simulations, we investigate universal rare events of critical $O(n)$ systems. Additionally, we explore the crossover from universal to nonuniversal behavior in PDF tails, extending Cramér's series to strongly correlated variables. Our findings highlight the universal and nonuniversal aspects of rare event statistics. We also discuss the ubiquity of this power-law corrections to the leading compressed-exponential decay in these tails in and out-of-equilibrium.



# 1 Introduction

The comprehension of rare events holds great significance in the study of complex systems encompassing diverse fields such as climate science, brain activity, societies, financial markets, and earthquakes. The occurrence of exceptional and dramatic phenomena arises from emergent behaviors within these systems. When they occur in large stochastic systems, these rare events can have universal characteristics. This is typically the case for systems exhibiting scaling, a situation encountered for systems that are close to a second-order phase transition or that are generically scale-invariant, i.e. without fine-tuning of any parameter, as in the Kardar-Parisi-Zhang (KPZ) equation describing interface growth. Predicting and analyzing such events is generally difficult because of the strong correlations between the degrees of freedom involved.

From an analytical point of view, the characterization of the rare events is contained in the tails of the probability distribution functions (PDF) $P(\hat{s} = s)$ of the normalized sum $\hat{s}$ of the stochastic variables of the system. Generically, the presence of strong correlations in scale invariant systems makes it necessary to use special techniques such as the Functional Renormalization Group (FRG) to obtain a complete characterization of the PDF and of its tail. Most of the time, it is therefore difficult to have fully controlled results concerning these rare events. When there is scale invariance, typically the leading behavior of the decay of the tails is a compressed exponential ruled by a critical exponent and is therefore not too difficult to obtain. For instance, for the $d$-dimensional Ising model, the leading behavior of the tail of the PDF is $\exp(-aL^d s^{\delta+1})$ where $a$ is a constant, $L$ the linear dimension of the system and $\delta$ the critical isotherm exponent [1]. However, this exponential decay can be accompanied by a nontrivial subleading term which is difficult to obtain, except when exact results are available.

A full understanding of these tails is important for at least three reasons. The first and obvious reason is conceptual: we want to fully characterize the statistics of the rare events. The second reason is related to the consistency of the different behavior of the PDF according to the value of its argument. For instance, for KPZ in $1+1$ dimension, there are different regimes depending on the behavior of the fluctuations of the height $H$ of the interface as a function of the time $t$. For the typical height fluctuations, $H$ behaves as $t^{1/3}$ and the PDF of these typical fluctuations is given by the Tracy-Widom distribution. Atypical large height fluctuations correspond to $H \sim O(t)$ and satisfy other distributions [2]. Obviously, these different behavior should match, the large field behavior of one distribution being the small field behaviors of the other. The matching between these different regimes has been proven for KPZ and it requires a detailed understanding of the tails of these distributions. The third reason is pragmatic: a quantitative fit of a PDF requires knowing it on the largest possible range which requires detailed knowledge of its tail, which has been argued to be mandatory for the Ising model in $d = 3$ [3].

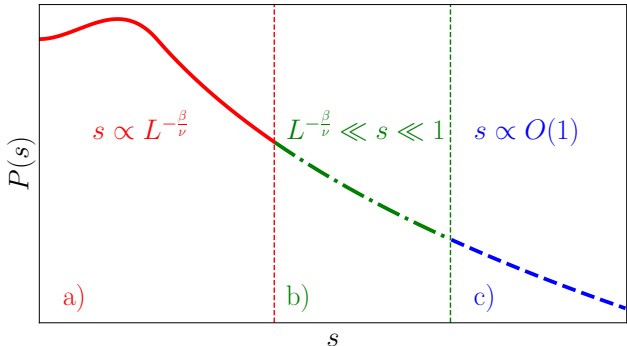

Figure 1: Schematic representation of different regimes of the probability distribution function of an $O(n)$ critical system. The regime a) is the scaling regime where the probability distribution is a universal function of $sL^{\beta/\nu}$. This regime corresponds to a generalization of the CLT to strongly correlated variables. The universal large deviation regime b) appears for $sL^{\beta/\nu} \gg 1$, where the PDF takes the form Eq. (1). This region is the main focus of the present work. Finally, regime c) is the non-universal large deviation regime. The cross-over from b) to c) is characterized by universal corrections to scaling multiplied by non-universal amplitudes, see Sec. 4.

For all the reasons mentioned above, the universal statistics of the rare events have been much studied in the last decades, especially for the Ising model close to criticality. For the models in the Ising universality class, it has been argued that a power-law correction to the leading exponential decay should be present [4], i.e. at large $s$

$$P_L(\hat{s} = s) \propto s^{\psi} e^{-aL^d s^{\delta+1}}, \tag{1}$$

with $\psi = \frac{\delta-1}{2}$ on heuristic ground [5] or assuming some analytic properties of the free energy [3,6–8]. This expression of the PDF has been argued in [8] to hold also out of equilibrium with $\psi$ being again $(\delta-1)/2$ for a one-component degree of freedom $x$, say a position in space, if the PDF is a scaling function of $x$ and $t$.

Our objectives in this article are twofold. We first want to show that Eq. (1) is most likely valid for $O(n)$ models with periodic boundary conditions with $\psi = n\frac{\delta-1}{2}$, by a set of different methods: perturbation theory, FRG, hierarchical Ising model, large $n$ limit, and Monte Carlo (MC) simulations. Although the methods and models are well-established, and we provide a brief review of each, our compilation of results on the power-law prefactor of the PDF and subsequent analysis constitutes, to the best of our knowledge, a novel contribution. We also argue that the existence of a power-law prefactor as in Eq. (1) with $\psi = n\frac{\delta-1}{2}$ is not necessarily present neither at nor out of equilibrium. We show it by considering the Ising model in $d = 3$ with free boundary conditions. Equation (1) is then invalid because a subleading power-law term corrects the leading $s^{\delta+1}$ term and hides the $s^{\psi}$ term. For out-of-equilibrium systems, using exact results for KPZ derived in [2], we show that the exponent $\psi$ is not necessarily $(\delta-1)/2$ which invalidates the argument put forward in [8].

Our second objective is to study the crossover between the universal tail of the PDF described by Eq. (1) which is valid for $s \sim L^{-\beta/\nu}$ with $\beta$ and $\nu$ respectively the order parameter exponent and the correlation length exponent, and the nonuniversal behavior of the PDF which holds for $s \gg L^{-\beta/\nu}$. For independent and identically distributed (iid) random variables $\hat{\sigma}_i$, this crossover which takes place for $\hat{s} = \sum_i \hat{\sigma}_i/L^d \sim 1$, is given by the Cramér's series. We argue that this Cramér's series can be generalized to the case of strongly correlated variables

and that it is given by a sum of contributions, each term of which corresponds to a correction-to-scaling exponent and its associated universal function. The non-universality of this series only appears in the amplitudes multiplying each of these contributions. Finally, this series has a finite radius of convergence, and beyond this radius, the PDF is fully nonuniversal, that is, is strongly dependent on the joint probability distribution of the $\hat{\sigma}_i$.

The manuscript is organized as follows. In Sec. 2 we recall the theory of large deviations and its connection to the Central Limit Theorem via Cramér's series for independent and weakly dependent variables. We then discuss how this picture is modified for strongly correlated variables in the context of second-order phase transitions. In Sec. 3, we characterize universal large deviations and show that Eq. (1) is obeyed for a variety of models. In Sec. 4, we discuss the connection between correction to scaling and Cramér's series, and we discuss the generality of our results in Sec. 5.

## 2  A short reminder on CLT and large deviations

### 2.1  Central limit theorem and Cramér's series for independent variables

For the sum of $N$ independent identically distributed (iid) random variables $\hat{\sigma}_i$, $\hat{S} = \sum_i \hat{\sigma}_i$, the Central Limit Theorem (CLT) and the Large Deviation Principle (LDP) allow for describing the typical fluctuations $\hat{S} \sim \sqrt{N}$ and large deviations $\hat{S} \sim N$ from the mean, respectively. (We assume that $\hat{\sigma}_i$ has zero mean and finite variance to simplify the discussion.) On the one hand, independently of the PDF of the $\hat{\sigma}_i$, the CLT implies that in the limit $N \to \infty$, the typical fluctuations of $\hat{S}$ are Gaussian, with standard deviation scaling as $\sqrt{N}$. On the other hand, the LDP asserts that for large deviations, $\hat{S}$ of order $N$, the PDF takes the form

$$P(\hat{S} = Ns) \simeq \sqrt{NI''(s)/2\pi}\, e^{-NI(s)}, \tag{2}$$

where the rate function $I(s)$ strongly depends on the probability distribution of $\hat{\sigma}_i$, i.e. it is non-universal in the language of critical systems. The derivation of this result, known as Cramér's theorem in the large deviation literature, is standard, see for instance [9]. It follows from a saddle-point approximation of the integral representation of the PDF

$$\begin{aligned} P(\hat{S} = Ns) &= \langle \delta(\hat{S} - Ns) \rangle \\ &= \int_{a-i\infty}^{a+i\infty} \frac{dh}{2i\pi} e^{-Nhs} \langle e^{h\hat{S}} \rangle, \end{aligned} \tag{3}$$

where the average $\langle \ldots \rangle$ is over the joint probability of the $\hat{\sigma}_i$. The integral over $h$ is performed on the Bromwich contour, i.e. along a vertical line $h = a$ in the complex plane. The real number $a$ is chosen so that the line $h = a$ lies to the right of all singularities. Notice that $\langle e^{h\hat{S}} \rangle$ is the moment generating function of $\hat{S}$, and $w(h) = N^{-1} \ln \langle e^{h\hat{S}} \rangle$ its cumulant generating function. For iid variables, we of course have that $w(h) = \ln \langle e^{h\hat{\sigma}_i} \rangle$, where the average is over $\hat{\sigma}_i$ only. Then

$$\begin{aligned} P(\hat{S} = Ns) &= \int_{a-i\infty}^{a+i\infty} \frac{dh}{2i\pi} e^{-N(hs-w(h))} \\ &\simeq \sqrt{N/2\pi w''(h^*)} e^{-N(h^*s-w(h^*))}, \end{aligned} \tag{4}$$

where we have performed a saddle-point approximation (including Gaussian fluctuations) in the limit $N \to \infty$, and $h^*$ is found as $\sup_{h \in \mathbb{R}}(hs - w(h))$ (note that the minimum of $hs - w(h)$ along the Bromwich contour is a maximum for $h$ real). Here, the Bromwich contour has been

deformed to go through its real saddle point, the existence of which is ensured by the fact that the PDF is real. Assuming that $w(h)$ is analytic, then $h^*$ is such that $w'(h^*) = s$. We introduce the average $m(h) = w'(h)$, and $U(m) = \sup_{h \in \mathbb{R}}(hm - w(h))$, which have a clear interpretation in statistical physics (see below). In the case of iid, we thus recover Eq. (2) with $I(s) = U(m = s)$, using the fact that $U''(m(h)) = w''(h)^{-1}$. Note that by construction $U(m)$ is always convex, while $I(s)$ needs not to be in general. Therefore, and this will be important below, the identification $I(s) = U(m = s)$ can only work in the regions where the rate function is convex.

In the present setting, the CLT can be reframed as

$$P(\hat{S} = \sqrt{N}\tilde{s}) \simeq \frac{e^{-I''(0)\tilde{s}^2/2}}{\sqrt{2\pi/I''(0)}}, \qquad (5)$$

for $\tilde{s}$ of order 1. The Gaussian distribution is universal (up to a non-universal "amplitude" $1/\sqrt{I''(0)}$ characterizing the typical fluctuations of $\hat{\sigma}_i$, i.e. the width of the PDF). CLT and LDP are related by noting that

$$P(\hat{S} = \sqrt{N}\tilde{s}) \simeq \frac{e^{-I''(0)\tilde{s}^2/2}}{\sqrt{2\pi/I''(0)}} e^{\frac{\tilde{s}^3}{\sqrt{N}}\lambda(\tilde{s}/\sqrt{N})}, \qquad (6)$$

for $\tilde{s} = o(\sqrt{N})$, i.e. for small deviations of $\hat{S}$ from its mean. Here $\lambda(z) = \sum_{k=0} a_k z^k$ is related to the so-called Cramér's series, which has a convergent series expansion around $z = 0$ corresponding to the series expansion of $I(s)$ with $s = \tilde{s}/\sqrt{N}$. The coefficients $a_k$ are related to the moments of the iid variables and are thus non-universal. Then $\lambda(z)$ plays the role of "finite size corrections" to the Gaussian distribution, with universal power-laws in $N$ but non-universal amplitudes. We refer to the mathematical literature for more rigorous statements, see e.g. [10, Chap. 8]. As the scale of $\tilde{s}$ increases to $O(\sqrt{N})$, the probability distribution crosses over into the fully non-universal regime. This happens because it becomes dominated by the Cramér's expansion, as it effectively reconstructs the rate function $I(s)$, which strongly depends on the microscopic distribution of the random variable.

## 2.2 Weakly dependent random variables

The above discussion can be straightforwardly generalized to dependent variables, where the joint probability distribution $\mathcal{P}[\hat{\sigma}]$ of the random variables does not factorize. This is for instance the case of the high-temperature phase of Ising spins $\hat{\sigma}_i = \pm 1$ on a $d$-dimensional hypercubic lattice of linear size $L$ ($N = L^d$) with nearest-neighbor interactions. Weak correlation amounts to $\langle \hat{S}^2 \rangle = N\chi$, with finite susceptibility $\chi$, which is ensured by the finite correlation length $\xi$. As the number of spins increases the PDF of the rescaled variables $\hat{S}/\sqrt{N}$ tends to a Gaussian: it is attracted to the (universal) high-temperature fixed point. In particular, the derivation presented above applies directly, as long as $L \gg \xi$ which ensures that $\lim_{N\to\infty} N^{-1} \ln\langle e^{h\hat{S}} \rangle$ is well defined and analytic for all $h$.

In this context, $w(h)$ is (minus) the Helmoltz free energy, while $U(m)$ is the Gibbs free energy, with $m = \langle \hat{S} \rangle/N$ the average magnetization. In the high-temperature phase, the rate function is convex, and $I(s) = U(m = s)$ for all $s$. This corresponds to the equivalence of ensembles in the thermodynamic limit, between a free energy $I(s)$ at fixed magnetization $s$ (canonical ensemble) and a free energy $U(m)$ at fixed average magnetization $m$ (grand canonical ensemble).

Large deviations are non-universal, depending on the shape of $I(s)$ at $s \sim 1$, strongly dependent on the microscopic distribution of the random variable (e.g. Ising vs soft spins). The Cramér's series in this case corresponds to correction to scaling to the high-temperature fixed point, with universal scaling form $\tilde{s}^{3+i}/N^{1+i}$, $i \in \mathbb{N}$, and non-universal prefactors (which depend on the derivatives of $I(s)$ at $s = 0$).

## 2.3 Strongly correlated variables

When the variables are strongly correlated, such as is the case close to a second-order phase transition, the CLT does not apply anymore. A signature is that the typical fluctuations of the variables scale differently than predicted by the CLT. The typical fluctuations of the normalized total spin $\hat{s} = \hat{S}/L^d$ at criticality are of order $L^{-(d-2+\eta)/2}$ instead of $L^{-d/2}$ (we use bold symbols for $O(n)$ spins). Here $\eta$ is the anomalous dimension of the field, and we will often use $\beta/\nu = (d-2+\eta)/2$ with $\beta$ and $\nu$ the magnetization and correlation length critical exponents respectively.

For $|\hat{s}|$ of order $L^{-\beta/\nu}$ the PDF of $\hat{s}$ takes the scaling form

$$P_L(\hat{s} = s) = L^{n\beta/\nu} p(s L^{\beta/\nu}), \tag{7}$$

where we used that the $O(n)$ symmetry implies that it only depends on $s = |s|$. Here $p(\tilde{s})$ is a $n$-dependent universal scaling function. The normalization of $p(\tilde{s})$ is such that $\int_0^\infty d\tilde{s}\, \tilde{s}^{n-1} p(\tilde{s}) = 1$ and $\int_0^\infty d\tilde{s}\, \tilde{s}^{n-1} \tilde{s}^2 p(\tilde{s}) = 1$. The second condition fixes the (non-universal) scale of the field and ensures that $p(\tilde{s})$ is fully universal (does not depend on non-universal amplitudes). It is highly non-Gaussian, with a shape that depends strongly on how the limits $T \to T_c$ and $L \to \infty$ are taken [11] (we will consider only the case $T = T_c$, $L \to \infty$ here for simplicity, unless stated otherwise), as well as the boundary conditions [12] (we assume periodic boundary conditions unless specified otherwise). However, the fact that $p(\tilde{s})$ is universal (for a given universality class) can be interpreted as a (non-rigorous) generalization of the CLT to strongly correlated variables (at least those corresponding to second-order phase transitions). This regime corresponds to the region a) of Fig. 1.

The critical PDF $p(\tilde{s})$ is typically non-monotonous for $\tilde{s} \propto O(1)$, as has been observed in simulations [6, 7, 12–14], perturbative and non-perturbative renormalization group analysis [11, 15–18]. This implies that the rate function $I(s)$ is non-convex for $s$ of order $L^{-\beta/\nu}$, and the relation $I(s) = U(m = s)$ breaks down. This is due to the fact that for $s \sim L^{-\beta/\nu}$, the typical magnetic field is of order $L^{-d+\beta/\nu}$ while the free energy scales as $w(\tilde{h} L^{-d+\beta/\nu}) = L^{-d} f(\tilde{h})$ for $\tilde{h}$ of order 1 (here $f(\tilde{h})$ is a universal scaling function). Thus, the exponent $L^d(sh - w(h))$ in the integral representation of the PDF is of order one (i.e. the factor $L^d$ disappears), and the saddle-point approximation breaks down.

On the other hand, for $L^{-\beta/\nu} \ll s \ll 1$, one expects to recover the thermodynamic limit behavior typical of critical scaling [1]

$$P_L(\hat{s} = s) \propto e^{-a L^d s^{\delta+1}}, \tag{8}$$

with $\delta = \frac{d+2-\eta}{d-2+\eta}$ the critical isotherm exponent and $a$ a constant. Note that since $(\delta + 1)\beta/\nu = d$, Eqs. (7) and (8) are consistent provided that $p(\tilde{s}) \propto e^{-\tilde{a}\tilde{s}^{\delta+1}}$ for $\tilde{s} \gg 1$. Here $\tilde{a}$ is universal and related to $a$ by a non-universal amplitude related to the scale of $s$. This behavior has been proven rigorously for the two-dimensional Ising model [19, 20] and for the hierarchical model [21], and is a natural consequence of the (functional) renormalization group [11]. It can be understood by realizing that in the thermodynamic limit $L \to \infty$ and $s$ fixed but not too large (i.e. much smaller than one), we can use the saddle-point approximation once again, using that $w(h) \propto h^{1+1/\delta}$ in this universal regime.

Note that the PDF in Eq. (8) takes a large deviation form, i.e. its logarithm scales with the volume, that is *universal*. On the contrary, for $s$ of order 1, the probability distribution is non-universal and depends on the microscopic details of the system. Therefore, contrary to what happens for iid variables, large deviations can be universal (if not too large) or non-universal, see regime b) and c) of Fig. 1. As we will discuss in Sec. 4, the equivalent of Cramèr's series that connects those two regimes are the finite-size effects associated with corrections to scaling.

Finally, let us give an argument for the $O(n)$ universality class that a better description of universal large deviation than Eq. (8) is Eq. (1), with $\psi = n\frac{\delta-1}{2}$. Since $L^{-\beta/\nu} \ll |\hat{s}| \ll 1$ corresponds to large fields where both the rate function $I(s)$ and the Gibbs free energy $U(m)$ are convex, the same saddle-point argument as above implies that

$$P_L(\hat{s} = s) \simeq (L^d U''(s)/2\pi)^{1/2}(L^d U'(s)/s2\pi)^{\frac{n-1}{2}} e^{-L^d U(s)}, \tag{9}$$

where the first prefactor comes from the longitudinal fluctuations with respect to $s$ and the second comes from the $n-1$ transverse fluctuations. Assuming no logarithm in $U(m)$ (which has not yet been proven so far) and scaling ($U(m) \propto m^{\delta+1}$ at large $m$) we obtain the prefactor $s^\psi$ of the Eq. (1), with $\psi = n\frac{\delta-1}{2}$, generalizing the Ising result to $O(n)$.

## 3 Universal large deviations

We now characterize the universal large deviations for a variety of models close to a second-order phase transition belonging to the $O(n)$ universality class, and show that they are consistent with Eq. (1) with $\psi = n\frac{\delta-1}{2}$.

In particular, we demonstrate that this relation holds true both for exact calculations (for the hierarchical model and at large $n$) or approximate ones (perturbative and functional RG), irrespective of the actual value of $\delta$ (exact or approximated). Some of the present models or approximation schemes might correspond to vanishing anomalous dimension $\eta$ (at lowest order of the expansion). However, we do not expect this to impact the validity of $\psi = n\frac{\delta-1}{2}$, in particular since in all cases studied here, we have a non-trivial exponent $\delta$ (i.e. non-mean-field) irrespective of the actual value of $\eta$. This is also confirmed by our Monte Carlo simulations in three dimensions, for which $\eta$ is finite.

### 3.1 Exactly solvable models

#### 3.1.1 Hierarchical model

The hierarchical model is one of the few models where explicit and rigorous results can be obtained at criticality. We refer to [22] for a review of the model and the derivations of the recursion relation of the PDF. The model describes a hierarchy of block-spins $S$ of size $N_k = 2^k$ with interaction strength $\left(\frac{c}{4}\right)^k$. The PDF $P_{(k)}(\tilde{s})$ of a block-spin at the $k$-th level of the hierarchy, with $\tilde{s} = \left(\frac{c}{4}\right)^{k/2} S$ the rescaled block-spin, obeys the recursion relation

$$P_{(k+1)}(\tilde{s}) \propto e^{\frac{\beta}{2}\tilde{s}^2} \int dx P_{(k)}\left(\frac{\tilde{s}}{\sqrt{c}} + x\right) P_{(k)}\left(\frac{\tilde{s}}{\sqrt{c}} - x\right). \tag{10}$$

While one cannot speak of the dimensionality of the hierarchical model, it is possible to make the connection with an effective dimensional $d$ as follows. Call $\ell_k^d = N_k$ the number of spins in a block-spin at level $k$. Then by analogy with a $d$-dimensional system, we write $\tilde{s} = \ell_k^{\frac{d-2+\eta}{2}} s = \ell_k^{\frac{-d-2+\eta}{2}} S$, or $\tilde{s} = N_k^{\frac{-d-2+\eta}{2d}} S$. Matching this expression with $\tilde{s} = \left(\frac{c}{4}\right)^{k/2} S$, we can link the parameter $c$ to the effective dimension of the system, $c = 2^{\frac{d-2+\eta}{d}}$. Note that because there is no real dimensionality of the system, one cannot disentangle the dimension $d$ and anomalous dimension $\eta$ in this case. It is then convenient to impose that $\eta$ here takes the value of the anomalous dimension of the Ising model in dimension $d$.

Reframed in this way, the critical behavior of the hierarchical model is very similar to that of the $d$-dimensional Ising model. At fixed $c$ and initial condition $P_{(0)}$, there is a transition at a critical $\beta$ for $c > 1$. For $c \geq \sqrt{2}$, corresponding to $d \geq 4$ assuming $\eta = 0$, the transition is

mean-field-like, and the fixed point of the recurrence relation is the once-unstable Gaussian. The Gaussian becomes twice-unstable for $c < \sqrt{2}$ and a new non-trivial fixed point emerges as $c$ crosses $\sqrt{2}$. One can even perform the equivalent to the epsilon expansion when $c$ is close to $\sqrt{2}$, i.e. $d = 4 - \epsilon$. For $c \in ]1, \sqrt{2}[$, the case on which we focus on, if the initial condition is properly fine-tuned, $P_{(k)}$ reaches asymptotically a once-unstable non-trivial fixed point $P_\star$, characterized by non-trivial critical exponents.

It is convenient to extract a Gaussian part from the probability and to introduce

$$g_{(k)}(\tilde{s}) = e^{A_\star \tilde{s}^2} P_{(k)}(\tilde{s}), \tag{11}$$

with $A_\star = \frac{\beta c}{2(2-c)}$. (The Gaussian PDF $P_\star = e^{-A_\star \tilde{s}^2}$ is a twice-unstable fixed point.) The fixed point equation for $g$ then reads

$$g_\star(\tilde{s}) \propto \int dx\, e^{-2A^\star x^2} g_\star\left(\frac{\tilde{s}}{\sqrt{c}} + x\right) g_\star\left(\frac{\tilde{s}}{\sqrt{c}} - x\right). \tag{12}$$

Let us now show that the critical PDF of the hierarchical model does take the form Eq. (1) in the critical rare events regime. A first simple argument goes as follows. Since the integral over $x$ is cut by the Gaussian weight, we expect that for sufficiently large $\tilde{s}$ the functions $g_\star$ (or more appropriately their logs) can be expanded in $x$. Keeping the leading term (i.e. neglecting their $x$ dependence), one obtains [23, 24]

$$g_\star(\tilde{s}) \propto g_\star\left(\frac{\tilde{s}}{\sqrt{c}}\right)^2, \tag{13}$$

which is solved by

$$g_\star(\tilde{s}) \propto e^{-a\tilde{s}^{\delta+1}}, \tag{14}$$

with $\delta + 1 = 2/\ln_2 c$, i.e. $\delta \in ]4, \infty[$ depending on $c$. This behavior has been demonstrated rigorously for $c = 2^{1/3}$ in [21]. Inserting Eq. (14) into Eq. (12), it is straightforward to see that the integral over $x$ generates a prefactor $\tilde{s}^{-\frac{\delta-1}{2}}$, which must be compensated for by requiring

$$g_\star(\tilde{s}) \propto \tilde{s}^{\frac{\delta-1}{2}} e^{-a\tilde{s}^{\delta+1}}. \tag{15}$$

We now give a more systematic analysis of the problem. Write $g_\star(\tilde{s}) = e^{-u_\star(\tilde{s})}$ and assume that for $\tilde{s} \gg 1$, $u_\star^{(n)}(\tilde{s}) \gg u_\star^{(n+1)}(\tilde{s})$ with $u_\star^{(n)}$ the $n$-th derivative of $u_\star$ (this assumption turns out to be self-consistent). Expanding in $x$ in the integrand of Eq. (12), and keeping the first two terms in the asymptotic expansion, we obtain (up to a constant)

$$u_\star(\tilde{s}) = 2u_\star(\tilde{s}/\sqrt{c}) + \frac{1}{2}\ln\left(2A^\star + u_\star^{(2)}(\tilde{s}/\sqrt{c})\right) + \cdots, \tag{16}$$

where the neglected terms are of order $u_\star^{(2n)}(\tilde{s}/\sqrt{c})/(u_\star^{(2)}(\tilde{s}/\sqrt{c}))^n$. At leading order we recover $u_\star(\tilde{s}) = 2u_\star(\tilde{s}/\sqrt{c})$, again solved by $u_\star(\tilde{s}) = a\tilde{s}^{\delta+1}$. This implies that $u_\star^{(2)}(\tilde{s}/\sqrt{c}) \propto \tilde{s}^{\delta-1}$ is much larger than $A^\star$. Keeping the leading term from the log, we find $u_\star(\tilde{s}) \simeq a\tilde{s}^{\delta+1} - \frac{\delta-1}{2}\ln\tilde{s}$ up to a constant, while the next term implies a subdominant power-law behavior $\tilde{s}^{-\delta+1}$. Note that the neglected terms in Eq. (16) are of order at most $\tilde{s}^{-\delta-1}$. The results obtained here are consistent with the rigorous large deviation analysis of [25].

### 3.1.2 Large $n$ limit

The large $n$ limit of the $O(n)$ model is another exactly solvable model, see [26] and [27, Chapt. 14] for a review of the derivation. The model is described by the Hamiltonian

$$\mathcal{H}[\hat{\boldsymbol{\phi}}] = \int_x \left(\frac{(\nabla\hat{\boldsymbol{\phi}})^2}{2} + V\left(\hat{\boldsymbol{\phi}}^2/2\right)\right), \tag{17}$$

with $V(x)$ is the potential, such that $V(nx)/n$ is independent of $n$, typically of the form

$$V(x) = r_0 x + \frac{u_0}{6n} x^2. \tag{18}$$

At some critical value $r_0$, there is a continuous phase transition between an ordered and a disordered phase for $d > 2$. Above the upper critical dimension $d = 4$, the phase transition is mean-field, while it is non-trivial for $2 < d < 4$, with critical exponents $\nu = 1/(d-2)$ and $\eta = 0$. This implies in particular $\beta = 1/2$ and $\delta + 1 = 2d/(d-2)$.

The PDF of the $O(n)$ model is defined by

$$P_L(\hat{s} = s) = \mathcal{N} \int \mathcal{D}\hat{\boldsymbol{\phi}}\, \delta(s - \hat{s}) \exp(-\mathcal{H}[\hat{\boldsymbol{\phi}}]), \tag{19}$$

with $\mathcal{N}$ a normalization constant, $\hat{s} = L^{-d} \int_x \hat{\boldsymbol{\phi}}(x)$. The delta-function can be exponentiated (see [28, 29] for a similar calculation using a different exponentiation of the delta-function), $\delta(z) \propto \lim_{M \to \infty} e^{-\frac{M^2}{2} z^2}$, such that

$$P_L(\hat{s} = s) = \lim_{M \to \infty} \mathcal{N}' \int \mathcal{D}\hat{\boldsymbol{\phi}}\, e^{-\mathcal{H}[\hat{\boldsymbol{\phi}}] - \frac{M^2}{2}(s - \hat{s})^2}. \tag{20}$$

Introducing two auxiliary fields $\lambda(x)$ and $\hat{\rho}(x)$ such that $1 = \int \mathcal{D}\lambda \mathcal{D}\hat{\rho} \exp\left\{-i \int_x \lambda\left(\frac{\hat{\phi}^2}{2} - \hat{\rho}\right)\right\}$, the PDF is rewritten as

$$P_L(s) = \lim_{M \to \infty} \mathcal{N}' \int \mathcal{D}\hat{\boldsymbol{\phi}}\mathcal{D}\lambda\mathcal{D}\hat{\rho}\, e^{-\int_x\left(\frac{(\nabla\hat{\phi})^2}{2} + i\lambda\frac{\hat{\phi}^2}{2}\right) - \int_x (V(\hat{\rho}) - i\lambda\hat{\rho}) - \frac{M^2}{2}(s - \hat{s})^2}. \tag{21}$$

Writing the field $\hat{\boldsymbol{\phi}} = (\hat{\sigma}, \hat{\boldsymbol{\pi}})$, with $\hat{\sigma}$ along the direction of $s$, and integrating out the $\hat{\boldsymbol{\pi}}$ fields, we finally obtain

$$P_L(s) = \lim_{M \to \infty} \mathcal{N}' \int \mathcal{D}\hat{\sigma}\mathcal{D}\lambda\mathcal{D}\hat{\rho}\, e^{-\mathcal{H}_{\text{eff}}[\hat{\sigma}, \lambda, \hat{\rho}]}, \tag{22}$$

with

$$\mathcal{H}_{\text{eff}}[\hat{\sigma}, \lambda, \hat{\rho}] = \int_x \left(\frac{(\nabla\hat{\sigma})^2}{2} + i\lambda\frac{\hat{\sigma}^2}{2}\right) + \int_x (V(\hat{\rho}) - i\lambda\hat{\rho})$$
$$+ \frac{M^2}{2}\left(L^{-d}\int_x (\hat{\sigma} - s)\right)^2 + \frac{n-1}{2}\text{Tr}\log(g_\pi^{-1}), \tag{23}$$

and the correlation function $g_\pi$ of the $\hat{\boldsymbol{\pi}}$-fields satisfying $(-\nabla^2 + i\lambda(x) + M^2)g_\pi(x, y) = \delta(x - y)$. Assuming that $\hat{\sigma} \sim \sqrt{n}$, the functional integral can be evaluated by a the saddle-point analysis as $n \to \infty$, and the PDF reads

$$P_L(s) = \lim_{M \to \infty} \mathcal{N}' e^{-\mathcal{H}_{\text{eff}}[\hat{\sigma}_0, \lambda_0, \hat{\rho}_0]}, \tag{24}$$

where $\hat{\sigma}_0, \lambda_0, \hat{\rho}_0$ minimize the effective Hamiltonian $\mathcal{H}_{\text{eff}}$. Assuming that the saddle is at constant field configurations, the limit $M \to \infty$ imposes $\hat{\sigma}_0(x) = s$, and we obtain

$$i\lambda_0 = V'(\hat{\rho}_0),$$
$$\frac{s^2}{2} = \hat{\rho}_0 - \frac{n}{2L^d}\sum_{q \neq 0} \frac{1}{q^2 + i\lambda_0}. \tag{25}$$

Writing $\log(P_L(s)) = -L^d I(\rho)$ with $\rho = s^2/2$, one shows that at the saddle-point, $i\lambda_0 = I'(\rho)$.

In the scaling regime, e.g. for $\rho$ small enough such that $I'(\rho) \ll u_0^{2/(4-d)}$ for the potential given in Eq. (18) (with the Ginzburg length $u_0^{-1/(4-d)}$ much smaller than $L$), we obtain the self-consistent equation

$$\rho = -n\Delta - \frac{n}{2L^d} \tilde{\sum}_{q \neq 0} \frac{1}{q^2 + I'(\rho)}, \tag{26}$$

where $\tilde{\sum}$ means the sum over momenta has been regularized at large momenta and $\Delta$ is the distance to the critical point ($\Delta > 0$ corresponding to the disordered phase in the thermodynamic limit). Following [26], this equation can be rewritten as

$$\rho = -n\Delta + \frac{n}{L^{d-2}} F_d\left(\frac{L^2 I'(\rho)}{4\pi}\right), \tag{27}$$

where

$$F_d(z) = -\frac{1}{2} \int_0^\infty \frac{du}{4\pi} \left(e^{-uz}(\vartheta^d(u) - 1) - u^{-d/2}\right), \tag{28}$$

with $\vartheta(u) = \sum_{k \in \mathbb{Z}} e^{-u\pi k^2}$ is a Jacobi theta function.

Looking for universal rare events at criticality ($\Delta = 0$) corresponds to $L^{-2\beta/\nu} \ll \rho/n \ll u_0^{\frac{d-2}{4-d}}$, with $\beta/\nu = (d-2)/2$ in large $n$, and $L^2 I'(\rho) \gg 1$. The large $z$ behavior of $F_d(z)$ reads

$$F_d(z) = A_d z^{\frac{d-2}{2}} + \frac{1}{8\pi z} + \mathcal{O}\left(e^{-2\sqrt{\pi z}}\right), \tag{29}$$

with $A_d = -\frac{\Gamma(1-d/2)}{8\pi} > 0$, where the first term corresponds to the result in the thermodynamic limit, while the second one comes from the subtraction of the $q = 0$ term in the sum, and is subdominant in the limit $L \to \infty$. Thus the self-consistent equation for its solution $I_\star$ reads

$$\rho/n \simeq A_d \left(\frac{I'_\star}{4\pi}\right)^{\frac{d-2}{2}} + \frac{1}{2L^d I'_\star}, \tag{30}$$

which is solved by

$$I'_\star(\rho) \simeq 4\pi \left(\frac{\rho}{nA_d}\right)^{\frac{2}{d-2}} - \frac{n}{L^d(d-2)\rho}. \tag{31}$$

Integrating with respect to $\rho$, we obtain

$$I_\star(\rho) \simeq c\rho^{\frac{d}{d-2}} - \frac{n}{L^d(d-2)} \ln(\rho), \tag{32}$$

up to a constant. Recalling that $\rho = s^2/2$, we thus obtain that for rare events $L^{-\beta/\nu} \ll s/\sqrt{n} \ll u_0^{\frac{d-2}{2(4-d)}}$,

$$P_L(s) \propto s^{n\frac{\delta-1}{2}} e^{-aL^d s^{\delta+1}}, \tag{33}$$

with $\delta = \frac{d+2}{d-2}$ in large $n$.

On the other hand, in the limit $\rho \gg u_0^{\frac{d-2}{4-d}}$, the universal term $F_d$ is subdominant and we recover $I(\rho) = V(\rho)$, corresponding to the non-universal regime of rare events,

$$P_L(s) \propto e^{-L^d V(s^2/2)}. \tag{34}$$

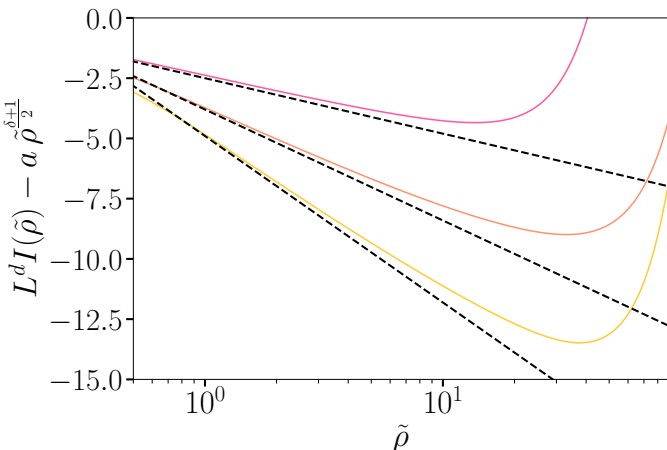

Figure 2: Rate function of the three-dimensional $O(n)$ model with its leading pow-erlaw behavior subtracted, $L^d I(\tilde\rho) - a\tilde\rho^{\frac{\delta+1}{2}}$, as a function of $\rho$, obtained from FRG for $n = 1, 2, 3$ (top to bottom). Here $\rho = s^2/2$ and $\tilde\rho = L^{2\beta/\nu}\rho$, with $2\beta/\nu = (d-2)$ and $\delta = 2d/(d-2)$ at LPA. The dashed lines correspond to $-n\frac{\delta-1}{4}\log(\tilde\rho)$ (note the log-scale of the abscissa).

## 3.2 Perturbative results in dimension $d = 4 - \epsilon$

The rate function at $T = T_c$ can also be computed in perturbation theory using the $\epsilon = 4 - d$ expansion, which reads [15, 18, 30]

$$
\begin{aligned}
L^d I(x) = {} & \frac{n+8}{9}\frac{2\pi^2}{\epsilon}x^4 + \pi^2 x^4\left(\gamma + \log 2\pi - \frac{3}{2} + \log(x^2)\right) + \frac{1}{2}\Delta_4\left(2x^2\right) \\
& + (n-1)\left[\frac{\pi^2}{9}x^4\left(\gamma + \log 2\pi - \frac{3}{2} + \log\left(\frac{x^2}{3}\right)\right) + \frac{1}{2}\Delta_4\left(\frac{2x^2}{3}\right)\right] + \mathcal{O}(\epsilon),
\end{aligned}
\tag{35}
$$

with $x = \sqrt{g_*}L^{\beta/\nu}s$ with $\beta/\nu = 1 + \mathcal{O}(\epsilon)$ and $g_* = \frac{3\epsilon}{n+8} + \mathcal{O}(\epsilon^2)$ is the fixed point value of the interaction to leading order in $\epsilon$. Here $\Delta_d(z) = \theta_d(z) - \theta_d(0)$ with

$$
\theta_d(z) = -\int_0^\infty ds\,\frac{e^{-sz}}{s}\left(\vartheta^d(s) - 1 - (1/s)^{d/2}\right),
\tag{36}
$$

is the integral of $F_d(z)$, up to a factor $4\pi$ and the subtraction of a term that diverges in $d = 4$. In particular, $\Delta_4(z) \simeq -\log(z)$ at large $z$.

At large field, $x \gg 1$, the leading behavior of the rate function is

$$
L^d I(x) \simeq \frac{n+8}{9}\frac{2\pi^2}{\epsilon}x^4\left(1 + \epsilon\log(x) + \mathcal{O}(\epsilon^2)\right),
\tag{37}
$$

which corresponds to the expected behavior $L^d I(x) \propto x^{\delta+1}$ with $\delta = 3 + \epsilon + \mathcal{O}(\epsilon^2)$, expanded to order $\epsilon$. This log behavior is an artifact of the $\epsilon$-expansion and can be dealt with using RG improvement to resum the large logs [15,18,31]. On the other hand, the contribution of $\Delta_4(x)$ at large $x$ gives a log correction $-n\log(x)$, which corresponds to the power-law prefactor $s^\psi$ with $\psi = n + \mathcal{O}(\epsilon)$ which is indeed equal to $n\frac{\delta-1}{2}$ to leading order. The calculation has been performed at two-loop recently for $n = 1$, and confirms this picture, giving $\delta = 3 + \epsilon + \frac{25}{54}\epsilon^2$ (as well known) and $\psi = 1 + \frac{\epsilon}{2} = \frac{\delta-1}{2}$ to first order in $\epsilon$ [32]. This suggests that the relationship $\psi = \frac{\delta-1}{2}$ holds order by order of the epsilon expansion.

### 3.3 Functional renormalization group

Recently, we have shown that the critical rate function of the Ising model can be computed from the FRG [11], see e.g. [33] for a review of FRG. Using the simplest non-trivial approximation, the so-called Local Potential Approximation (LPA), we were able to compute the PDF at criticality, in good agreement with Monte Carlo simulations. This is easily generalized to the $O(n)$ model [29]. We implement Wilson's idea of integration of the microscopic degrees of freedom by modifying the Hamiltonian in Eq. (19), $\mathcal{H}[\hat{\boldsymbol{\phi}}] \to \mathcal{H}[\hat{\boldsymbol{\phi}}] + \Delta H_k[\hat{\boldsymbol{\phi}}]$. One then obtains an equation for a scale-dependent rate function $I_k$. Following the standard procedure of FRG [33], we choose $\Delta H_k[\hat{\boldsymbol{\phi}}] = \frac{1}{2L^d} \sum_q R_k(\boldsymbol{q})\hat{\boldsymbol{\phi}}(\boldsymbol{q}).\hat{\boldsymbol{\phi}}(-\boldsymbol{q})$, where $k$ is the RG momentum scale and $R_k(\boldsymbol{q})$ is a regulator function that freezes the low wavenumber fluctuations ($q \ll k$) while leaving unchanged the high wavenumber modes ($q \gg k$). It is chosen such that: (i) when $k$ is of order of the inverse lattice spacing, $R_k(\boldsymbol{q}) \to \infty$, and all fluctuations are frozen; (ii) $R_{k=0}(\boldsymbol{q}) \equiv 0$, all fluctuations are integrated out, and $P_L(\boldsymbol{s}) \propto e^{-L^d I_{k=0}(\boldsymbol{s}^2/2)}$.

The flow equation at LPA reads

$$\partial_k I_k = \frac{1}{2L^d} \sum_{\boldsymbol{q} \neq 0} \partial_k R_k(\boldsymbol{q}) \left( \frac{1}{\boldsymbol{q}^2 + R_k(\boldsymbol{q}) + I'_k + 2\rho I''_k} + \frac{n-1}{\boldsymbol{q}^2 + R_k(\boldsymbol{q}) + I'_k} \right). \tag{38}$$

In practice, we use the method described in [11] to numerically solve the flow equation and obtain the critical PDF for the $O(n)$ universality classes. See however Appendix A for a discussion of the technical subtleties specific to the study of the universal large deviations and not addressed in [11]. The LPA implies a vanishing anomalous dimension, and thus we should obtain a compressed exponential tail with $\delta + 1 = \frac{2d}{d-2}$ and a power-law prefactor with $\psi = n\frac{2}{d-2}$. Note that the LPA is exact in the large $n$ limit [34] and we recover the results discussed above in this limit.

Fig. 2 shows the rate functions of the $O(n)$ model where the leading power-law behavior $as^{\delta+1}$ is subtracted, in $d = 3$ for $n = 1, 2, 3$. We observe a behavior consistent with a subleading logarithmic term (appearing as a straight line in log-linear scale), with prefactor $n\frac{\delta-1}{2}$. At large field, we find a deviation from this behavior, which we ascribe to the numerical resolution of the flow equation (App. A). In particular, increasing the resolution of the grid used to numerically integrate the flow pushes this deviation to larger and larger fields.

### 3.4 Monte Carlo simulations of the 3D Ising model

We now proceed to show that there is a power-law prefactor in the PDF of the $3d$ Ising model on the cubic lattice with periodic boundary conditions. For this purpose, we use Monte Carlo simulations based on a specially modified version of the Swendsen-Wang (SW) cluster algorithm [35], similar in spirit to that of [36, 37].

SW cluster algorithm is a very efficient tool for simulations of the critical Ising model [38]. One step of the algorithm to get from one spin configuration to the next goes as follows: it first connects parallel spins into $n_{\mathcal{C}}$ clusters (with $n_{\mathcal{C}}$ a random variable). Then all spins of a given cluster are flipped with 50% probability, giving rise to a new spin configuration. Calling $\mathcal{S}_a = \pm 1$ the new direction of the spins of cluster $\mathcal{C}_a$ (made of $|\mathcal{C}_a|$ spins), the total magnetization after that step is then $M = \sum_{a=1}^{n_{\mathcal{C}}} \mathcal{S}_a |\mathcal{C}_a|$. Note that for a given cluster configuration $\{\mathcal{C}_a\}$, a given spin configuration is just one instance of $2^{n_{\mathcal{C}}}$ equally probable configurations (corresponding to the $2^{n_{\mathcal{C}}}$ possible values of $\{\mathcal{S}_a\}$). Therefore, an improved estimator to increase the statistics of the magnetization configurations is to take into account the $2^{n_{\mathcal{C}}}$ possible values of $\sum_{a=1}^{n_{\mathcal{C}}} \mathcal{S}_a |\mathcal{C}_a|$ (with corresponding weights).

In [36,37], an analytic method for such purpose was proposed for the quantum Heisenberg model. Here, we follow a different route, using the fact that most clusters are of very small

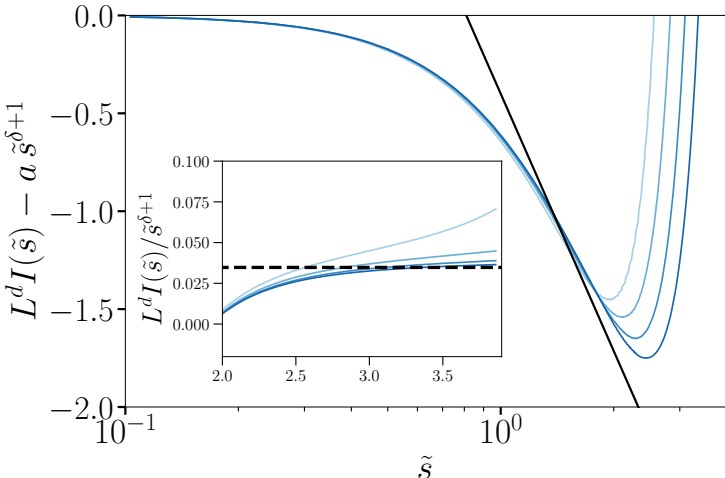

Figure 3: Rate function with its leading power law behavior subtracted, $L^d I(\tilde{s}) - a\tilde{s}^{\delta+1}$, as a function of $\tilde{s} = L^{\beta/\nu}s$, obtained from Monte Carlo simulations of the 3D Ising model of size $L = 16, 32, 64, 128$ (from light to dark blue) at criticality. The black line corresponds to $-\frac{\delta-1}{2}\log(\tilde{s})$ (note the log-scale of the abscissa). Inset: $I(\tilde{s})/\tilde{s}^{\delta+1}$ as a function of $\tilde{s}$ (linear scale). The dashed line corresponds to the constant $a \simeq 0.034$ extrapolated to infinite system size.

size, meaning that the sum over a typical configuration of the $\mathcal{S}_a$ of such clusters will average out to zero by the law of large numbers.[1] In particular, a configuration where most of those $\mathcal{S}_a$ points in the same direction will have a negligible weight and can be ignored. We, therefore, choose to sample exactly the orientation of the $k$ largest clusters (with $k$ fixed) and choose randomly the orientations of the $n_\mathcal{C} - k$ other clusters. Each configuration has a weight of $2^{-k}$. Our estimator is in principle less optimal than that of [36,37], though much better than a naive one considering only one orientation of the $n_\mathcal{C}$ clusters, but works very well for the present purpose.

In practice, we use the SW algorithm to construct the clusters, and a variation of the Hoshen-Kopelman method [40] to identify all the clusters for a given configuration. We typically generate $10^7$ cluster configurations. We then compute the magnetization for all possible orientations of the $k = 10$ largest clusters and update the PDF accordingly. To sample the tail of the distribution, we also introduce an external magnetic field to bias the system to larger than typical magnetization, using the ghost spin construction [35]. We then use multi-histogram reweighting to combine the data at various magnetic fields at zero field [41]. This allows us to probe the PDF to extremely rare events with probability as low as $e^{-200}$.

The results for the $3d$ Ising model with periodic boundary conditions are given in Fig. 3. As for the FRG results, we have subtracted the leading powerlaw behavior from the rate function, see App. A for details. We recall that in this case, $\beta/\nu \simeq 0.518149$ and $\delta \simeq 5.78984$ [42]. The figure shows conclusively the logarithmic correction (corresponding to a power-law prefactor for the PDF). However, determining the exponent $\psi$ is extremely sensitive to finite size effects which are still apparent for $L = 128$ (see Appendix A).

---

[1]At criticality, the average number $N_l$ of clusters of size $l$ obeys the scaling law $N_l = L^d l^{-\tau} f(l/L^{d_F})$, with $\tau = 1 + d/d_F$ and fractal dimension $d_F = \frac{d+2-\eta}{2}$, see e.g. [39]. There are thus an extensive number of small clusters, which contribute to the magnetization per site as a Gaussian variable of zero mean and standard deviation $\sim L^{-d/2}$. These contributions do not need to be taken into account, in the sense that after binning of the magnetization data, with a bin size that is a fraction of the typical magnetization $L^{-(d-2+\eta)/2}$, all these contributions fall into the same bin.

This leads us to comment on the strong finite-size effects observed in the universal rare events regime. As discussed above, this regime corresponds to $L^{-\beta/\nu} \ll s \ll 1$. Note that for the maximum size that we have, $L = 128$, $L^{-\beta/\nu} \sim 0.08$ and we do not even have a range of one decade in $s$ to observe this regime. The situation is even worse in $d = 2$, where the power-law is very strong since $\delta + 1 = 16$, and $\beta/\nu = 1/8$. This indicates that it is almost impossible to be in the universal rare event regime, since $L^{-\beta/\nu} \simeq 0.08$ even for $L = 10^9$. This casts doubts on the analyses performed on much smaller sizes in previous MC calculations for Ising $2d$ [7, 43–46].

## 4 Non-universal large deviations

We finally address how the RG allows us to understand how to relate universal and non-universal large deviations by generalizing the concept of the Cramérs' series, see also [47] for an early discussion about the connection between large deviation and RG. We discuss the Ising case here to simplify the notations, for which $|s| \leq 1$, without loss of generality.

Standard RG arguments imply that the rate function $I(s)$ takes a scaling form

$$I(s) = L^{-d}\tilde{I}_\star(sL^{\beta/\nu}),$$

for $s$ small enough and with $\tilde{I}_\star$ a universal function. We know that (at least in $d = 3$), the rate function is somewhat similar to the fixed point effective potential $\tilde{U}_\star$ of the FRG [11]. Furthermore, there are corrections to scaling which are of the form $\sum_m a_m L^{-\omega_m}\delta\tilde{I}_m(sL^{\beta/\nu})$, where the sum is over irrelevant perturbation with critical exponent $\omega_m > 0$.

By analogy with the connection between the fixed point potential and the rate function, we expect that the corrections to scaling $\delta\tilde{I}_m$ take a form similar to that of the irrelevant perturbations $\delta\tilde{u}_m$ to the fixed point with eigenvalue $\omega_m$. It is important to note that

$$\delta\tilde{u}_m(\tilde{\phi}) \propto c_m\tilde{\phi}^{(d+\omega_m)\nu/\beta},$$

at large field, while $\tilde{U}_\star(\tilde{\phi}) \sim c_\star\tilde{\phi}^{d\nu/\beta}$ for $\tilde{\phi} \to \infty$. Note however that $\delta\tilde{I}_m$ cannot be equal to $\delta\tilde{u}_m$ (or $\tilde{I}_\star$ to $\tilde{U}_\star$) since the former is universal while the latter depends on the RG scheme (e.g. the regulator function $R_k$ in FRG).

Thus, we predict that the rate function behaves for small enough $s$ as

$$I(s) = L^{-d}\tilde{I}_\star(sL^{\beta/\nu}) + \sum_m a_m L^{-d-\omega_m}\delta\tilde{I}_m(sL^{\beta/\nu}). \tag{39}$$

Let us stress here that the functional forms of $\tilde{I}_\star$ and $\delta\tilde{I}_m$, as well as $\omega_m$, are universal (i.e. described by the Wilson-Fisher fixed point) up to a non-universal amplitude associated with a characteristic scale of the random variables $\hat{\sigma}$. All other microscopic details associated with the joint probability distribution $\mathcal{P}[\hat{\sigma}]$ are encoded in $a_m$.

For large enough $L$, the PDF takes the form

$$P_L(\hat{s} = s) \simeq e^{-\tilde{I}_\star(sL^{\beta/\nu}) - \sum_m a_m L^{-\omega_m}\delta\tilde{I}_m(sL^{\beta/\nu})}. \tag{40}$$

We see that the typical fluctuations of $\hat{s}$ are of order $L^{-\beta/\nu} = L^{-(d-2+\eta)/2}$, instead of the standard $L^{-d/2}$ for iid variables, i.e. they are stronger by a factor $L^{1-\eta}$. Furthermore, we see that $\tilde{I}_\star(\tilde{s})$ does play the role of the universal distribution function of this generalized CLT, while $\sum_m a_m L^{-\omega_m}\delta\tilde{I}_m(\tilde{s})$ is a generalization of Cramér's series.

Much in the same way that the CLT breaks down for $N^{-1/2}\sum_i \hat{\sigma}_i$ of order $\sqrt{N}$, we find that the generalized CLT breaks down for $sL^{\beta/\nu} = \mathcal{O}(L^{\beta/\nu})$ (i.e. $s$ of order 1). Indeed, using

the large field behavior of the fixed point solution and its eigenperturbations, we find that in this regime

$$\tilde{I}_\star(sL^{\beta/\nu}) + \sum_m a_m L^{-\omega_m} \delta\tilde{I}_m(sL^{\beta/\nu}) \simeq L^d s^{d\nu/\beta}\left(c_\star + \sum_m a_m c_m s^{\omega_m \nu/\beta}\right), \qquad (41)$$

which shows that for $s$ of order 1, all "corrections" are of the same order and the expansion breaks down. Therefore, to see the universal feature of the tail of the PDF (in particular, the expected stretched exponential decay $\exp(-c_\star L^d s^{d\nu/\beta})$), one needs to be in the regime $L^{-\beta/\nu} \ll s \ll 1$.

All these aspects can be seen explicitly in the large $n$ limit, as we show now. If the system size is sufficiently large such that the finite-size corrections are negligible, we have seen in Sec. 3.1.2 that the rate function takes a universal form $I_\star$, solution of Eq. (26) at $\Delta = 0$. Then $\tilde{I}_\star$ defined above is just $L^d I_\star$.

To compute the correction to scaling, we restart from Eq. (25), which we can rewrite as

$$\rho = \frac{n}{L^{d-2}} F_d\left(\frac{L^2 I'(\rho)}{4\pi}\right) - \sum_{m \geq 2} m\, a_m (I'(\rho))^{m-1}, \qquad (42)$$

where we assume $\Delta = 0$ and the series $\sum_{m \geq 2} m\, a_m (I'(\rho))^{m-1}$ comes from the inversion of $I' = V'(\hat{\rho}_0)$ in Eq. (25) (the factor $-m$ and the power $m-1$ are chosen for later convenience). The amplitudes $a_m$ are non-universal and depend on the potential $V$. For instance, $a_m = -\delta_{m,2}\frac{3n}{2u_0}$ for the potential in Eq. (18). Assuming that $I = I_\star + \delta I$, using that

$$\rho = \frac{n}{L^{d-2}} F_d\left(\frac{L^2 I'_\star(\rho)}{4\pi}\right), \qquad (43)$$

we have

$$\frac{n}{4\pi L^{d-4}} F'_d\left(\frac{L^2 I'_\star(\rho)}{4\pi}\right) \delta I'(\rho) = \sum_{m \geq 2} m\, a_m (I'_\star(\rho))^{m-1}, \qquad (44)$$

where we have neglected higher order terms in $\delta I'$ and neglected subdominant terms in the scaling limit (e.g. $\delta I'/u_0$ compared to $\delta I'/L^{d-4}$). Furthermore, using that

$$I''_\star(\rho) \frac{n}{4\pi L^{d-4}} F'_d\left(\frac{L^2 I'_\star(\rho)}{4\pi}\right) = 1, \qquad (45)$$

we obtain $\delta I'(\rho) = \sum_{m \geq 2} m\, a_m (I'_\star(\rho))^{m-1} I''_\star(\rho)$, which implies

$$\delta I(\rho) = \sum_{m \geq 2} a_m (I'_\star(\rho))^m. \qquad (46)$$

Finally, using the fact that $I'_\star(\rho) = L^{-2} G_d(L^{d-2}\rho)$ with $G_d(\tilde{\rho}) \propto \tilde{I}'_\star(\tilde{\rho})$ a universal function of $\tilde{\rho} = L^{d-2}\rho = L^{2\beta/\nu}\rho$, we obtain that

$$L^d I(\rho) = \tilde{I}_\star(\tilde{\rho}) + \sum_{m \geq 2} a_m L^{-(2m-d)} G_d^m(\tilde{\rho}), \qquad (47)$$

from which we recover Eq. (40) with $\delta\tilde{I}_m(\tilde{\rho}) = G_d^m(\tilde{\rho})$ and $\omega_m = 2m - d$, which are indeed the correct critical exponents for the irrelevant perturbation to the Wilson-Fisher fixed point in large $n$ [34,48]. The large field behavior of $G_d(\tilde{\rho})$ is proportional to $\tilde{\rho}^{2/(d-2)}$, see Eq. (29), which implies that

$$\delta\tilde{I}_m(\tilde{\rho}) \propto \tilde{\rho}^{(d+\omega_m)\nu/2\beta}, \qquad (48)$$

in agreement with Eq. (41).

To finish this discussion, we can comment on the basin of attraction of this Generalized CLT. Focusing on the Ising universality class, we expect that a huge manifold in theory space of co-dimension 1 should be attracted to this universal distribution. In particular, assume that we know one model (say a $\hat{\phi}^4$ theory) that can be fine-tuned to criticality. Then we expect that we can smoothly modify the initial distribution while still being critical as long as one parameter is fine-tuned. While it is surely possible to modify the initial Boltzmann weight in such a way that the critical point disappears (say, by making the transition first order), we expect the basin of attraction to occupy a large part of the relevant domain of theory space.

For the three-dimensional Ising universality class and provided there is a unique fixed point associated with its critical behavior –which is commonly accepted– the parameter space at the phase transition, which is of codimension one in the full parameter space, is divided into two parts: the space where the transition is second order (II) and the space where it is first order (I). In I, the correlation length is finite at the transition and the system is therefore weakly correlated: the CLT applies under the standard form. In II, the RG flow is attracted towards the Wilson-Fisher FP and the rate function is nontrivial as well as its finite size corrections, Eq. (40). Thus, the basin of attraction of the GCLT is huge and corresponds to all models displaying a continuous phase transition belonging to the Ising universality class. The exception to the rule above is the border between I and II, which is of codimension 2 in the full parameter space. It is associated with multicritical behavior. Generically, on this multicritical hypersurface, the long-distance behavior is tricritical which is driven by the Gaussian fixed point in $d = 3$. This hypersurface has itself a boundary which is therefore of codimension three in the full parameter space where the behavior is quadricritical, also driven by the Gaussian fixed point in $d = 3$. The process never stops and there are infinitely many multicritical behaviors associated with attractive hypersurfaces of higher and higher codimensions. Notice that in $d = 2$ and for the Ising model, all multicritical behaviors are associated with nontrivial fixed points that are all different and thus must show a nontrivial PDF.

## 5 Discussion and conclusion

We have shown that in critical systems at equilibrium, rare events are described by a large deviation principle, having both a universal and non-universal regime. This is in contrast with weakly dependent or independent variables, for which rare events are described by a non-universal rate function. The universal regime is described by Eq. (1), with exponent $\psi = n\frac{\delta-1}{2}$, as explicitly shown in a variety of models at a second-order phase transition. The transition to the non-universal regime is described by finite-size correction to scaling, and characterized by the universal critical exponents corresponding to irrelevant perturbations of the fixed point describing the transition. This is the equivalent of Cramér's series for strongly correlated variables (at least when described by a Wilson-Fisher-like fixed point).

An important question concerns the generality of the results presented here. It has been argued in [8] that Eq. (1) with exponent $\psi = \frac{\delta-1}{2}$ (for one-component degree of freedom) also holds generically for out-of-equilibrium systems presenting anomalous diffusion. While the general argument presented in [8] is flawed, see Appendix B, it is also possible to find counter-examples where $\psi \neq \frac{\delta-1}{2}$. One such example is the PDF of the fluctuation height $H$ at time $t$ of the KPZ universality class. For typical fluctuations, $H \sim t^{1/3}$ and the PDF takes the form

$$P_\beta(H, t) \simeq t^{-1/3} f_\beta(H t^{-1/3}), \tag{49}$$

where $\beta = 1$ ($\beta = 2$) corresponds to the flat (droplet) initial condition and $f_\beta(z)$ is the Tracy-Widom distribution, see [2] and its supplementary materials for details. For large deviations,

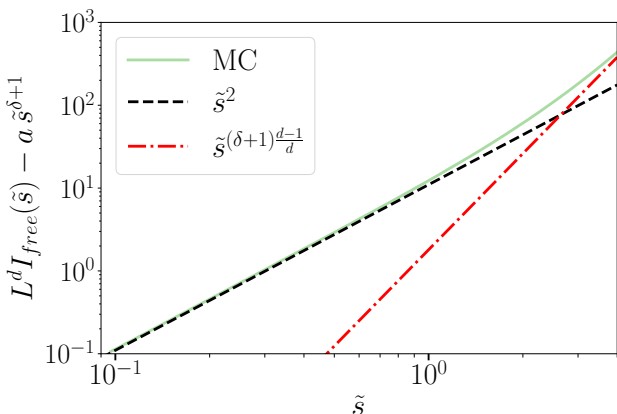

Figure 4: Rate function obtained from Monte Carlo simulations for the 3d Ising with $L = 64$ at $T_c$, but with free boundary conditions. Note that we have subtracted the same leading contribution $a\tilde{s}^{\delta+1}$ that we found for periodic boundary conditions (shown in Fig. 3). The dashed line corresponds to a quadratic behavior at small magnetization, while the dotted-dashed line corresponds to a surface correction term.

$Ht^{-1/3} \gg 1$, the Tracy-Widom distribution takes the asymptotic form

$$f_\beta(z) \propto z^{(2-3\beta)/4} e^{-\frac{2\beta}{3}z^{3/2}}, \tag{50}$$

from which we read $\delta = 1/2$ and $\psi = \frac{2-3\beta}{4}$. While for $\beta = 1$, we indeed have $\psi = -\frac{1}{4} = \frac{\delta-1}{2}$, this is not the case for $\beta = 2$ where $\psi = -1$. Therefore, while there is indeed a power-law prefactor in front of the universal compressed exponential term, the two powers need not be generically related to each other.

Coming back to critical systems at equilibrium, it is possible that the power-law prefactor may become difficult to observe if corrections to the leading behavior are stronger than the prefactor itself. To illustrate this, we present two examples.

If the system is slightly out of criticality, with $t = (T - T_c)/T_c \neq 0$ a relevant perturbation with negative eigenvalue $-1/\nu$, Eq. (39) is modified by an additional term (assuming $a_m = 0$ for simplicity)

$$L^d I(s) \approx \tilde{I}_\star(sL^{\beta/\nu}) + atL^{1/\nu}\delta\tilde{I}_\nu(sL^{\beta/\nu}). \tag{51}$$

Here $a$ is a non-universal amplitude that can be fixed by trading $at$ with $\xi_\infty \propto t^{-\nu}$, a diverging length scale of the infinite system, corresponding for instance to the correlation length in the disordered phase. Then $atL^{1/\nu} \to (L/\xi_\infty)^{1/\nu}$ and $I(s)$ written in terms of $\zeta = L/\xi_\infty$ and $sL^{\beta/\nu}$ is universal. In the scaling regime, $t \to 0$ and $L \to \infty$ (implying that irrelevant perturbations disappear) keeping $\zeta$ constant, the shape of the rate function is modified by $\delta\tilde{I}_\nu$, which behaves as $(sL^{\beta/\nu})^{(d+1/\nu)\nu/\beta}$ for $L^{-\beta/\nu} \ll s \ll 1$. It can therefore hide the powerlaw prefactor of the PDF if $\zeta$ is large enough.

Another possibility is a critical system with free boundary conditions, where the leading bulk term in the PDF, $L^d s^{\delta+1}$, might be corrected by subdominant but scaling surface term $\propto L^{d-1}s^{y_s}$. From the condition that this term obeys scaling, we find $y_s = (\delta+1)\frac{d-1}{d}$. This would modify the leading behavior of Eq. (8) in terms of the scaling variable $\tilde{s} = sL^{\beta/\nu}$ to

$$P_{L,f}(\hat{s} = s) \propto e^{-a\tilde{s}^{\delta+1} - b\tilde{s}^{(\delta+1)(d-1)/d} + \cdots}. \tag{52}$$

It is thus clear that in the region of rare events, the surface term would be far more important than a power law prefactor (which might nevertheless be there). We expect such a surface term to be present in a critical system with free boundary conditions. Figure 4 shows the

dimensionless rate function of the 3d Ising model with free boundary conditions, obtained from Monte Carlo simulations at $L = 64$. Note that the same leading behavior $a\tilde{s}^{\delta+1}$ as that in Fig. 3 has been subtracted, since we do not expect the bulk coefficient to be modified. We see that after a region where the rate function appears quadratic, it crosses over into a region that could be compatible with a surface term. In comparison with Fig. 3, we see that there is little chance of seeing a logarithmic behavior unless the surface term as well is removed, which is quite a formidable task. We do not doubt that analogous relevant examples could also be found out of equilibrium.

# Acknowledgments

AR and BD wish to thank the Institute of Physics for its hospitality, and BD and IB for that of the Université de Lille, where part of this work was done.

**Author contributions**   AR and IB conceived the research. All authors contributed equally to its writing. AR and IB performed the calculations and simulations.

**Funding information**   AR and IB are supported by the Croatian Science fund project HRZZ-IP-10-2022-9423 (I.Balog), an IEA CNRS project (A.Rançon), and by the "PHC COGITO" program (project number: 49149VE) funded by the French Ministry for Europe and Foreign Affairs, the French Ministry for Higher Education and Research, and The Croatian Ministry of Science and Education. IB wishes to acknowledge the support of the INFaR and FrustKor projects financed by the EU through the National Recovery and Resilience Plan (NRRP) 2021-2026.

# A   Extraction of the $\psi$ exponent numerically

In Sections 3.3 and 3.4, we show that the correction to the leading order behavior of the rate function is consistent with a logarithmic behavior, corresponding to a power-law prefactor in the PDF.

Determining the critical exponent $\psi$ remains challenging in numerical analyses of the rate function, whether obtained from solving the partial differential equation (FRG) or through simulations at finite sizes (MC). We discuss in this appendix these aspects in more detail.

## A.1   Extracting $\psi$ in FRG

First of all consider the solution of Eq. (38). Here, we used the exponential regulator

$$R_k(\boldsymbol{q}) = \alpha k^2 e^{-q^2/k^2},$$

with $\alpha \simeq 4.65$ corresponding approximately to the optimized (point of least dependence of the critical exponent on the regulator) value for all cases of $n$, the number of components of spin, that we consider in the present work. The numerical resolution of this problem was considered in detail in [11], from which we summarize the main steps. If one is interested in the universal scaling function $L^d I$, one can start the flow from a fixed point initial condition, at some initial scale $k_*$, corresponding to the solution of a dimensionless version of Eq. (38) in the thermodynamic limit. For a large $L \gg k_*^{-1}$, the flow is initially virtually vanishing. However as $k$ decreases, the flow starts differing from the thermodynamic-limit flow, and the flow essentially terminates for $kL \propto \mathcal{O}(1)$.

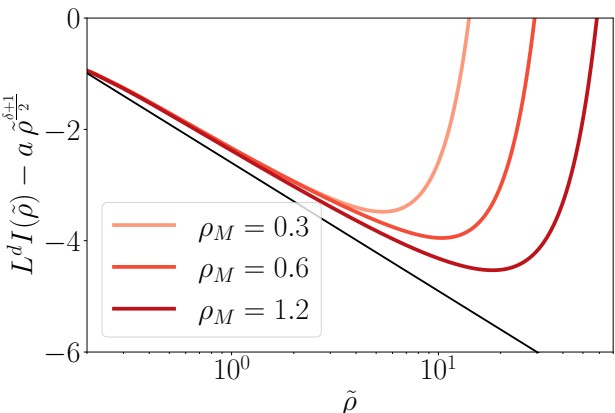

Figure 5: Subleading behavior of the rate function as a function of $\tilde{\rho} = L^{2\beta/\nu}\rho$ from FRG for $n = 1$ for various maximum ranges of the field $\rho_M$, keeping the grid mesh $\Delta\rho = 0.00075$ fixed. The black line shows the expected $-\frac{\delta-1}{2}\log\tilde{\rho}$. Increasing the $\rho_M$ allows for seeing the log behavior for larger and larger fields.

The $\rho$ dependence of the rate function is discretized on a grid, with mesh $\Delta\rho$ and maximum range $\rho_M$. We use grid parameters that are sufficient to describe the initial condition correctly. We run the flow in terms of dimensionless quantities (using for instance the variable $\rho k^{-2\beta/\nu}$, with $\beta/\nu = (d-2)/2$ at LPA, down to an RG scale $k_d$ (typically $4L^{-1}$–$10L^{-1}$), before switching to dimensionful quantities. Note that this means that the maximum value of $\rho$ has been shrunk by a factor of typically $L^{-2\beta/\nu}$, but ensures that the grid is fine enough to capture the behavior of the rate function for field values of order $L^{-2\beta/\nu}$. However, this also implies that to capture correctly the tail of the rate function, we need to start with a big enough range, and increasing it allows for recovering larger and larger sections of the tail.

From Eq. (1), we expect the rate function to behave as $\tilde{I}(\tilde{s}) \approx a\tilde{s}^{\delta+1} - \psi\ln(\tilde{s})$ for large enough $\tilde{s} = L^{\beta/\nu}s = L^{\beta/\nu}\sqrt{2\rho}$. We see that recovering the logarithmic tail on top of the leading power-law behavior requires determining the rate function to high precision. Typically the relative magnitude of the logarithmic term compared to the leading power-law behavior is $10^{-5}$ for $\tilde{s} \approx 10$.

Extending the range where the logarithmic behavior is seen can be achieved by increasing the size of the grid in $\rho$, i.e. increasing $\rho_M$. It is illustrated in Fig. 5. Furthermore, for a given $\rho_M$, the range can be extended by refining the mesh $\Delta\rho$ as seen in Fig. 6. The results presented here are for $n = 1$ but are representative. We also found that discretizing the derivatives following the recommendation of [49] also improves the large field behavior.

Assuming that the PDF behaves as Eq. (1) at large enough fields, writing (recall that $\rho = s^2/2$)

$$P_L(s) \propto e^{-L^d I(s)}$$
$$\propto e^{-\tilde{I}(\tilde{s})}, \tag{A.1}$$

the exponent $\psi$ can be recovered in principle from the numerical data by computing the estimator

$$e(\tilde{s}) = \frac{\tilde{s}^{\delta+2}}{(\delta+1)\ln(\tilde{s})-1}\frac{d}{d\tilde{s}}\left(\frac{\tilde{I}(\tilde{s})}{\tilde{s}^{\delta+1}}\right). \tag{A.2}$$

Indeed, $e(\tilde{s}) \to \psi$ for $\tilde{s}$ large enough if Eq. (1) is obeyed. The behavior of $e(\tilde{s})$ does not depend on $\rho_M$ as long as it is large enough, but it depends considerably on the mesh of the grid $\Delta\rho$, as seen in Fig. 7.

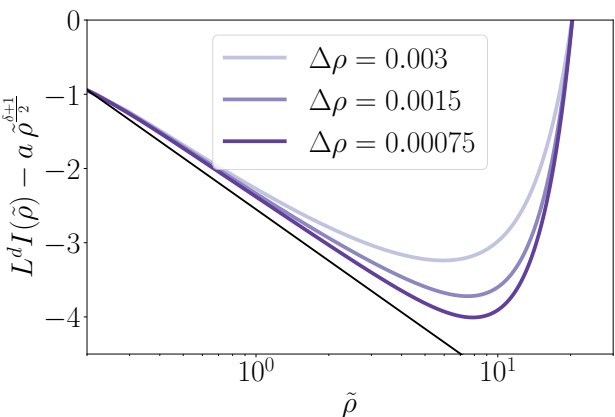

Figure 6: Subleading behavior of the rate function as a function of $\tilde{\rho} = L^{2\beta/\nu}\rho$ from FRG for $n = 1$ for various grid mesh of the field $\Delta\rho$, keeping the maximum range $\rho_M = 0.6$ fixed. The black line shows the expected $-\frac{\delta-1}{2}\log\tilde{\rho}$. Decreasing the $\Delta\rho$ allows for seeing the log behavior for larger and larger fields.

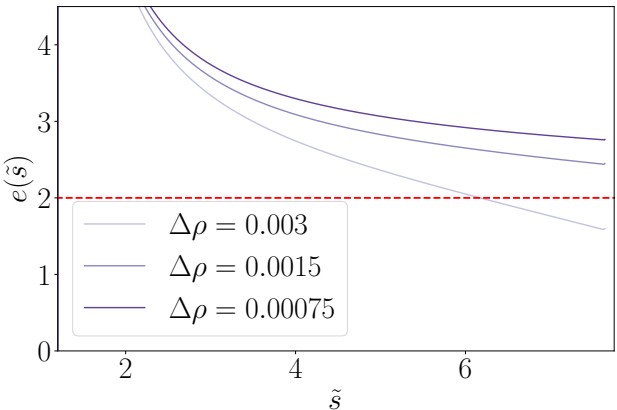

Figure 7: Estimator $e(\tilde{s})$, Eq. (A.2), as a function of $\tilde{s}$ for FRG at LPA and $n = 1$. Same data as in Fig. 6. Depending on the mesh size $\Delta\rho$, the estimator can overshoot the predicted plateau at $\psi = \frac{\delta-1}{2}$ (equal to 2 at LPA, shown as red dashed line). Decreasing $\Delta\rho$ improves the behavior of $e(\tilde{s})$.

## A.2 Extracting $\psi$ in MC

When we consider how well the exponent $\psi$ is captured from the Monte Carlo data, the challenges are different than in FRG determination. Here we are limited by the maximal $L$ that can be reasonably studied with high enough statistics. The range in which the logarithmic correction to the rate function can be observed in principle is for $L^{-\frac{\beta}{\nu}} \ll s \ll 1$. We see that even with our largest lattice $L = 128$ in $3d$, $L^{-\frac{\beta}{\nu}} \approx 0.08$, it is quite hard to achieve this regime.

We show in the inset of Fig. 3 that the leading power-law behavior $a_L s^{\delta+1}$ is sensible to finite-size corrections as $a_L$ has an $L$-dependence. We have extrapolated its value in the thermodynamic limit $a = \lim_{L\to\infty} a_L$ to subtract $as^{\delta+1}$ from the rate function. Note that in this range of field, the leading behavior is of the order of 200 while the correction is of order 1. Fig. 8 shows $e(\tilde{s})$, defined in Eq. (A.2), as determined from the Monte Carlo data. We observe that while there is a minimum, it is still far from $\psi = \frac{\delta-1}{2}$ due to finite size effects, even for $L = 128$.

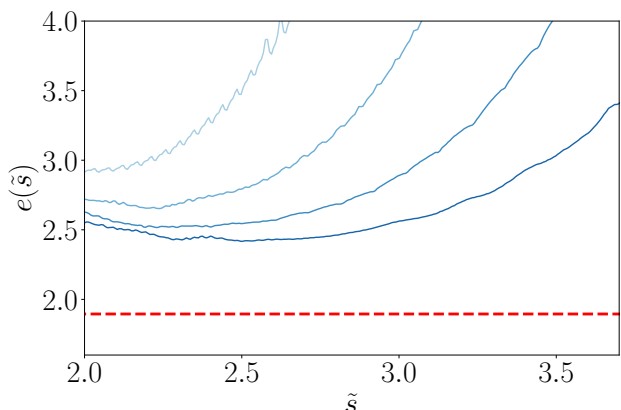

Figure 8: Estimator $e(\tilde{s})$, Eq. (A.2), as a function of $\tilde{s}$ from MC data for $L = 16, 32, 64, 128$ from light to dark blue. The dashed line corresponds to $\psi = \frac{\delta-1}{2}$. Finite-size corrections are still rather strong even for $L = 128$, while the maximum range accessible in $\tilde{s}$ is also rather limited and might not be yet in the deep universal rare event regime.

## B  Flaw in the argument of Ref. [8]

We summarize the argument of Ref. [8] to relate $\psi$ and $\delta$ and show why it is flawed. We also provide an explicit toy model to illustrate our point.

Ref. [8] argues the generating function

$$G(\lambda, t) = \int dx e^{\lambda x} p(x, t), \tag{B.1}$$

for scaling systems, $p(x,t) = t^{-\nu} f(x t^{-\nu})$, must be well defined for $t \to \infty$ and $\lambda \to 0$, which implies that if $p(z)$ is a stretched exponential with power-law $z^{\delta+1}$, then it must be of the form $z^{\psi} e^{-a z^{\delta+1}}$ with $\psi$ fixed to be equal to $\frac{\delta-1}{2}$. (They only consider one-component degrees of freedom, i.e. $n = 1$.)

The argument goes as follows. Using the change of variable $z = x t^{-\nu}$,

$$G(\lambda, t) = \int dz e^{\lambda t^{\nu} z} p(z), \tag{B.2}$$

and performing a saddle-point approximation, they find that if $p(z) \sim z^{\psi} e^{-a z^{\delta+1}}$ for a priori arbitrary $\psi$, then

$$\log G(\lambda, t) \simeq \lambda t^{\nu} \bar{z} - a \bar{z}^{\delta+1} + \frac{2\psi + 1 - \delta}{2} \log \bar{z} + \cdots, \tag{B.3}$$

where $\bar{z}$ satisfy the saddle-point condition $\bar{z} \propto (\lambda t^{\nu})^{1/\delta}$. Note that $\lambda t^{\nu} \bar{z} \sim \bar{z}^{\delta+1} \sim (\lambda t^{\nu})^{(\delta+1)/\delta}$.

They then argue that "The term $\propto \log \bar{z}$ is the only one which actually allows us to split the $\lambda$ and $t$ dependencies into the sum of two separate terms. Therefore its presence would introduce a logarithmic singular dependence on $\lambda$ in the whole $t$-independent part of $\log G$, implying a divergence for $\lambda = 0$. For such reason, this dependence should be dropped by the above choice of $[\psi = \frac{\delta-1}{2}]$." The flaw in this argument is that the presence of this logarithmic term does not imply a logarithmic divergence at $\lambda = 0$. Indeed, the saddle-point approximation assumes that the product $\lambda t^{\nu}$ is large. Thus one cannot simply take the limit $\lambda \to 0$ in

Eq. (B.3) without carefully taking the limit $t \to \infty$ (note that the real object of interest is $\frac{1}{t} \log G(\lambda, t)$ which is well defined in the limit $t \to \infty$ at fixed $\lambda$ provided $\delta = \frac{\nu}{1-\nu}$).

Thus, while it is true that in the limit $t \to \infty$, $\frac{1}{t} \log G(\lambda, t) \to \lambda^{1/\nu}$ can have non-analytic derivatives at $\lambda = 0$, it is not true that the exponent $\psi$ must be equal to $\frac{\delta-1}{2}$ to prevent a logarithmic (non-physical) divergence at $\lambda = 0$.

This is easily exemplified with the following toy model. Choose $p(z) = e^{-z^4}/2\Gamma(5/4)$, with $\Gamma(z)$ the Gamma function, corresponding to $\delta = 3$, $\nu = 3/4$ and $\psi = 0$. The generating function can be computed exactly in terms of hypergeometric functions,

$$G(\lambda, t) = {}_0F_2\left(; \frac{1}{2}, \frac{3}{4}; \frac{t^3\lambda^4}{256}\right) + \frac{\lambda^2 t^{3/2}\Gamma\left(\frac{3}{4}\right){}_0F_2\left(; \frac{5}{4}, \frac{3}{2}; \frac{t^3\lambda^4}{256}\right)}{8\,\Gamma\left(\frac{5}{4}\right)} . \tag{B.4}$$

Note that $G(0, t) = 1$ for all $t$ by normalization of $p(z)$, while

$$\lim_{t \to \infty} \frac{1}{t} \log G(\lambda, t) = \frac{3\lambda^{4/3}}{2^{8/3}} , \tag{B.5}$$

where the limit is taken at fixed $\lambda$. In the limit $t^{3/4}\lambda \gg 1$, we get

$$\log G(\lambda, t) \simeq \frac{3}{2^{8/3}}\lambda^{4/3}t - \log\left(\lambda^{1/3}t^{1/4}\right) + \cdots . \tag{B.6}$$

The leading term can be rewritten as $(\lambda t^{3/4})^{4/3} = (\lambda t^\nu)^{(\delta+1)/\delta}$ while the second reads

$$-\log((\lambda t^{3/4})^{1/3} = \frac{2\psi + 1 - \delta}{2}\log((\lambda t^\nu)^{1/\delta}), \tag{B.7}$$

in agreement with saddle-point calculation Eq. (B.3).

Therefore, the logarithmic term in Eq. (B.3) needs not to vanish in this regime (which would imply $\psi = \frac{\delta-1}{2}$, in contradiction with our choice of $p(z)$) for the generating function to be well defined at $\lambda = 0$, as asserted in Ref. [8].

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
