# Peer review of "Universal and non-universal large deviations in critical systems"

_SciPost Physics, doi:SciPost Phys. 18, 119 (2025)_

## Round 1 · Referee Report · Anonymous (Referee 1) · 2024-12-5

Strengths
Weaknesses
Report
Requested changes
-
One thing I have noted is that the exponents used in the analysis, beta/nu, delta, eta depend uniquely on the eigenvalue exponent y_h related to the field variable. I think this is because of the particular choice of order parameter fluctuations at T=T_c and small field. If one studied fluctuations of the enthalpy, H=U-Nhm, even at T=T_c, the critical scaling would involve y_t as well as y_h. Or, for the order parameter, working at zero field, with reduced temperature t would (I think) introduce the second exponent, y_t through the critical exponent nu. Could this be done ? Is it of interest ? Or does it just add complexity without content ?
-
I was a bit confused about some of the universality classes appearing in the analysis. In particular for the hierarchical model, I was expecting a tree calculation to yield mean field critical exponents but the expression seems to predict delta=5. Is that correct ? It is quite close to the D=3 value but I don't see dimensionality in the hierarchy. In addition to referencing [22] it would be worthwhile discussing this in the text.
-
Similarly, it would be useful to be reminded what one can expect in the large n limit. Naively I was again expecting mean field exponents, but we have 2beta/nu=d-1 and delta=2d/(d-2). References would be useful. These expressions predict unexpected results for d=4.
-
I appreciated the discussion of finite size effects and the consequences for entering the universal large deviation regime in d=2 and d=3. Does this mean that experimentally the regime might just be accessible in d=3 ? What about systems like helium-4 where the lambda transition can be approached with exquisite precision and where finite size effects such as the critical Casimir force can be measured ?
Recommendation
Publish (easily meets expectations and criteria for this Journal; among top 50%)

---

## Round 1 · Referee Report · Anonymous (Referee 2) · 2024-12-10

Report
The paper studies the probability distribution functions of O(n) models at their critical points. It compares the results obtained with different approaches and discusses the corrections to the leading behavior. The topic is old but relevant. There has recently been notable progress on this problem from the RG theory point of view.
The manuscript is written in a very reader-friendly style, which I perceived as a "mixture" of a short review and a research paper. I like, appreciate, and support this way of writing, but I insist that the revised manuscript makes a more sharp distinction between the previous and the new results. At the present point, it is not quite clear from the reading (at least for me), which results presented here are new and which appeared in literature before ( Refs. 11, 18 in particular, but also earlier papers).
In addition to this, I have the following points and questions:
- The abstract strongly says that the results: "challenge existing assumptions about power-law corrections...". If I am correct, the relevant reference to this sentence appears only in the discussion section. The first part of this discussion of Sec.5 only invoked previous results (out of equilibrium), the second one (relating to MC of 3d Ising) is short and rather soft or speculative. The authors should also say more explicitly and clearly what "assumptions" they mean and discuss the references. It is also unusual that a result very important for the paper is mentioned only in the concluding section (and is also rather speculative). In contrast, the result sections strongly confirm the expected behavior, but only for situations/approximations that have mean-field features (see the point below).
- I got the impression that (except for MC) the calculations of the paper have vanishing anomalous dimensions (due to either approximation or the properties of the model). In particular this seems to imply (quite unsatisfactorily) that the tail is the same for all values of n, which is not true in general. In consequence the impact of n only appears at the level of the correction. The important quantity beta/nu is also always the same at the level of the carried out calculations.
I do not see why the role of \eta should not be discussed, and also computed with functional RG or the perturbative approach (even at low levels of accuracy). I also see it as a bit logically tense: the paper studies correction factors with approximations which do not capture the correct leading behavior (which in my understanding involves \eta and depends on the value of n).
- It might be very interesting to discuss d=2 (and n=1). It seems like none of the present approaches can be applied to this case. Is this right? I understand that the motivation for such study can be limited due to the difficulty of reaching the universal regime in MC (but this difficulty occurs also in d=3).
- What is meant by "CLT breaks down"? A mathematical theorem cannot break down. I understand that the theorem assumptions are not met. But which assumptions? Similarly: what are the assumptions and statement of what the authors call "generalised CLT"?
- There is imbalance in detailing technical points of distinct methods. I propose to move most of Sec.3.4 (the technical part) to the appendix.
Requested changes
See the report -\ eta in particular.
Recommendation
Ask for major revision

---

## Round 1 · Referee Report · Anonymous (Referee 3) · 2024-12-15

Report
In this paper, the authors study large deviations in critical systems. While they mainly focus on a situation at equilibrium (namely the O(n) model and its variants), they also comment on some non-equilibrium situations, mainly the Kardar-Parisi-Zhang (KPZ) equation in 1+1 dimension. One of the motivation of the paper is the power-law correction (with corresponding exponent \psi) to the leading stretched exponential tail that describes the (typical) fluctuations of the magnetisation in these systems. The value of this exponent was conjectured to be $\psi = (\delta-1)/2$ where $\delta$ is the isotherm exponent. Furthermore, in Ref. [1] this conjecture was extended to non-equilibrium systems. The second focus of this paper is to explore the crossover from this typical stretched exponential behavior to the large deviation regime (which they call Cramer's regime).
Concerning the first point, the authors present different approaches (hierarchical model, mean-field limit, functional renormalization group, perturbative approach in $d=4-\epsilon$ and numerical simulations) which all suggest that Eq. (1) with indeed the exponent $\psi$ as predicted (multiplied by $n$ for a $n$-component system), confirming the conjectures stated in previous works. However they also point out, using previous exact results for the KPZ equation that the value of the exponent $\psi$ is different. Concerning the large deviations, they actually show that there are two regimes of large deviations: a first one for moderate values of the parameter $s$ -- which to a large extent is universal in the renormalization group sense -- and a second one for very large values of $n$ which is non universal.
The paper is well written, physically sound and present interesting and timely results. I believe that they will be of interest to the statistical physics community working on large deviations and/or critical phenomena. Therefore, I would like to recommend the publication of the present manuscript in SciPost. I only have minor comments that the authors may address in a revised version of their manuscript.
1) On line 40-44 the authors discuss an algebraic day but instead mention a stretched exponential behavior right after.
2) On line 43, when they discuss the Ising model, they should say that in this case 's' is the global magnetisation of the system.
3) In Eq. (6) the denominator should be \sqrt{2\pi/I''(0)}
4) In Eq. (7), on the left hand side, this should be $s$ and not s.
5) Below Eq. (B6) the text needs to formatted properly.
Recommendation
Publish (easily meets expectations and criteria for this Journal; among top 50%)

---

## Round 2 · Referee Report · Anonymous (Referee 3) · 2025-1-21

Report

In this revised version of the manuscript, the authors have satisfactorily taken into account my comments and made the appropriate modification. Hence, I can now recommend the publication of this manuscript.

Recommendation

Publish (easily meets expectations and criteria for this Journal; among top 50%)

---

## Round 2 · Referee Report · Anonymous (Referee 1) · 2025-1-26

Strengths

The paper is well written and informative and the authors have successfully dealt with all referees comments.

Weaknesses

None remaining

Report

The paper is well written and informative and the authors have successfully dealt with all referees comments.

Recommendation

Publish (surpasses expectations and criteria for this Journal; among top 10%)

---

## Round 2 · Referee Report · Anonymous (Referee 2) · 2025-1-31

Report

The Authors have made several clarifications and amendments to the manuscript. Lack of \eta presents a drawback in my opinion, but, despite this, the paper content is interesting, relevant, broad and original. I fully recommend its publication in SciPost Physics.

Recommendation

Publish (surpasses expectations and criteria for this Journal; among top 10%)

---

## Round 2 · Author Response

Dear Editors and Referees,

We resubmit our article and wish to take this opportunity to thank all the referees for their pertinent and insightful comments which we believe have improved considerably the article. We proceed by responding to all of the referee's comments and listing the changes made to the article to accommodate them.

Referee 1 (Report 2)

We thank the Referee for their review. We have carefully addressed the Referee’s points in the new version of the manuscript and in our replies below.

1) "The manuscript is written in a very reader-friendly style, which I perceived as a ”mixture” of a short review and a research paper. I like, appreciate, and support this way of writing, but I insist that the revised manuscript makes a more sharp distinction between the previous and the new results. At the present point, it is not quite clear from the reading (at least for me), which results presented here are new and which appeared in literature before ( Refs. 11, 18 in particular, but also earlier papers)."

We thank the Referee for their appreciation of our writing style. While Section 2 (the reminder on CLT) is indeed a short review on CLT and Cramer’s series and of the critical PDF, the research of the manuscript is new. Of course, we are using, in part, existing calculations in the literature (1-loop and large N calculation, as well as results on the hierarchical model), and thus need to review the main feature of these models. However, the core of the article is on the powerlaw prefactor to the PDF, which has not been investigated in the given references. So while our article compiles many different results in the literature (and we also perform additional calculations using Monte Carlo and FRG, not performed in 11), our interpretations and conclusions are new. We have nevertheless followed the Referee’s advice and made this point clearer in the new version of the manuscript.

2) " The abstract strongly says that the results: ”challenge existing assumptions about power-law corrections...”. If I am correct, the relevant reference to this sentence appears only in the discussion section. The first part of this discussion of Sec.5 only invoked previous results (out of equilibrium), the second one (relating to MC of 3d Ising) is short and rather soft or speculative. The authors should also say more explicitly and clearly what ”assumptions” they mean and discuss the references. It is also unusual that a result very important for the paper is mentioned only in the concluding section (and is also rather speculative). In contrast, the result sections strongly confirm the expected behavior, but only for situations/approximations that have mean-field features (see the point below)."

We agree with the Referee that the wording in the abstract might not have been appropriate, and we have toned down this sentence, replacing it by ”We also discuss the ubiquity of this power-law corrections to the leading compressed-exponential decay in these tails in and out-of-equilibrium.”. It is then less striking that the corresponding discussion appears in the concluding section, and that it is at times more speculative.

However, as also discussed in the response to other comments, the result section does confirm the expected behavior, but NOT only for situations/approximations that have mean-field features. While some have η = 0, this does not imply that they are mean-field, since e.g. the hierarchical model and the O(n) model in large n are both described by a non-trivial fixed point at the transition.

3) " I got the impression that (except for MC) the calculations of the paper have vanishing anomalous dimensions (due to either approximation or the properties of the model). In particular this seems to imply (quite unsatisfactorily) that the tail is the same for all values of n, which is not true in general. In consequence the impact of n only appears at the level of the correction. The important quantity beta/nu is also always the same at the level of the carried out calculations. I do not see why the role of η should not be discussed, and also computed with functional RG or the perturbative approach (even at low levels of accuracy). I also see it as a bit logically tense: the paper studies correction factors with approximations which do not capture the correct leading behavior (which in my understanding involves η and depends on the value of n). "

One of the major goals of the manuscript is to verify the relationship between the exponents ψ and δ (i.e. ψ = n (δ−1)/2 ). What we show is that this relationship is obeyed even after using approximations. We have added a comment at the beginning of the Sec. 3 where we explain that this is tested regardless of the approximation. The role of η that the referee points out amounts here to the precision of the exponent δ = (d+2−η)/(d−2+η) (and correspondingly ψ) as determined by the calculation. Thus, using models or approximations for which η = 0 does not affect our main conclusion. A collaborator of ours has recently performed the calculation of the rate function at order 2 in epsilon, at which the anomalous dimension is non-zero, and found that the relation between ψ and δ holds at that order, suggesting that it holds order by order in epsilon. Finally, we note that, as explained in answering to point 2 of the Referee 3, while there is no η in the case of the ierarchical model, the exponent δ can be changed at will, and the exponent ψ changes accordingly. We have clarified this point.

4) " It might be very interesting to discuss d=2 (and n=1). It seems like none of the present approaches can be applied to this case. Is this right? I understand that the motivation for such study can be limited due to the difficulty of reaching the universal regime in MC (but this difficulty occurs also in d=3). "

The case of the 2D Ising model is indeed interesting. While the FRG cannot address this case at the Local Potential Approximation (as used here), it is possible to perform a more complex calculation (at the next order in the derivative expansion), which is known to work for Ising 2D. We are currently writing a manuscript on how to adapt the derivative 2 expansion for the rate function, in which we also discuss how well the method fares. These issues however have no bearing on the relation between exponents ψ and δ that we discuss here and would not change anything. While it is in principle possible to use NPRG to study the effect in d = 2, we did not do it here because it would be impossible to corroborate the result by MC as explained in Sec. 3.4.

5) " What is meant by ”CLT breaks down”? A mathematical theorem cannot break down. I understand that the theorem assumptions are not met. But which assumptions? Similarly: what are the assumptions and statement of what the authors call ”generalised CLT”?"

We agree with the Referee that our phrasing was confusing. As the Referee notes, what we meant is that some conditions for the CLT to hold are not met for critical systems. In this case, it is the divergence of the variance of √(Ŝ/ N) , that implies that the CLT to does apply (if it is finite, then the system is weakly correlated and (a version of) the CLT holds, as discussed in Sec. IIB). Unfortunately, there is no rigorous definition of a generalized CLT for critical systems, so we cannot give the corresponding assumptions and a precise rigorous statement of such a theorem. However, the RG gives a heuristic understanding of what a generalized CLT should look like.

We have corrected this in the text.

6) "There is imbalance in detailing technical points of distinct methods. I propose to move most of Sec.3.4 (the technical part) to the appendix."

We have carefully considered the Referee’s suggestion, but have decided not to implement it. While technical, there are some original ideas in the way of performing simulations that were crucial for going so deep into the rare events region. We feel that these details would be lost in an appendix.

---

## Round 2 · List of Changes

List of changes (Referee 1):

  • Rewriting of the last sentence of the abstract.
  • Line 72: New sentence related to point 1.
  • Lines 165 and 181: change of wording corresponding to point 5.
  • Lines 220-227: Added a paragraph corresponding to point 3.
  • Lines 328-331: Mention of the results at order 2 in epsilon for the perturbative calculation, see point 3. -Line 348: added sentence related to point 1.

List of changes (Referee 2):

1) "On line 40-44 the authors discuss an algebraic day but instead mention a stretched exponential behavior right after."

The phrasing was indeed unclear, we meant the powerlaw behavior inside the exponential. This has been clarified.

2) "On line 43, when they discuss the Ising model, they should say that in this case ’s’ is the global magnetisation of the system."

Now s is defined before being used in this expression.

3)" In Eq. (6) the denominator should be √2π/I ′′ (0)"

We thank the Referee for pointing out the typo, it is corrected.

4)" In Eq. (7), on the left hand side, this should be s and not s."

There were some issues in the formatting of bold symbols using the Scipost template. This has been corrected in all equations.

5) "Below Eq. (B6) the text needs to formatted properly."

This has been corrected.

List of changes (Referee 3):

  • Lines 235 to 251: additional details on the hierarchical model to address point 2.
  • Lines 275 to 281: additional references for the large n limit and details of the critical behavior to address point 3.
  • Lines 506 to 518: additional discussion on the effect of temperature away from criticality, addressing point 1.

---

## Editorial Decision

published